# Same Pandemic Yet Different COVID-19 Vaccination Roll-Out Rates in Two Small European Islands: A Comparison between Cyprus and Malta

**DOI:** 10.3390/healthcare10020222

**Published:** 2022-01-24

**Authors:** Sarah Cuschieri, Amalia Hatziyianni, Marios Kantaris, Antonis Kontemeniotis, Mamas Theodorou, Elena Pallari

**Affiliations:** 1Department of Anatomy, Faculty of Medicine and Surgery, University of Malta, MSD 2080 Msida, Malta; sarah.cuschieri@um.edu.mt; 2Ammochostos General Hospital, 25 Christou Kkeli, Paralimni 5310, Cyprus; hatziyianniamalia@gmail.com; 3Health & Social Services Research Centre, American University of Cyprus, Larnaca 6019, Cyprus; marioskantaris@healthresearch.cy; 4Ministry of Health, Prodromou 1, Nicosia 1148, Cyprus; akontemeniotis@gmail.com; 5Health Policy, Open University of Cyprus, Latsia 2220, Cyprus; m.theodorou@ouc.ac.cy; 6MRC Clinical Trials Unit, University College London, London WC1V 6LJ, UK

**Keywords:** COVID-19, mass vaccination, vaccine hesitancy, mortality, morbidity, Cyprus, Malta

## Abstract

A mass vaccination strategy is estimated to be the long-term solution to control COVID-19. Different European countries have committed to vaccination strategies with variable population inoculation rates. We sought to investigate the extent to which the COVID-19 vaccination strategies, inoculation rate, and COVID-19 outcome differ between Cyprus and Malta. Data were obtained from the Ministry of Health websites and COVID-19 dashboards, while vaccination data were obtained from the European Centre for Disease Prevention and Control until mid-June, 2021. Comparative assessments were performed between the two countries using Microsoft^®^ Excel for Mac, Version 16.54. Both islands took part in the European Union’s advanced purchase agreement and received their first batch of vaccines on 27 December 2020. The positivity rate and mortality between December and June differs between the two countries (average positivity rate Cyprus 1.34, Malta 3.37 *p* ≤ 0.01; average mortality Cyprus 7.29, Malta 9.68 *p* ≤ 0.01). Both the positivity rate and mortality for Cyprus declined due to strict public health measures and vaccination roll-out in early January (positivity rate by 95% and mortality by 58%). In contrast, for Malta, there was a sharp increase (64% *p* ≤ 0.01) with almost no public health restrictions in place and soaring cases during the Christmas and Carnival period until March, when lockdown measures were re-introduced. A distinctive difference between Cyprus and Malta in positivity rate (14 per 100,000 population; *p* ≤ 0.01) can also be observed between January and mid-April 2021. However, from April onwards it is evident that the positivity rate and mortality decline (positivity rate Cyprus by 82%, Malta by 95%; mortality Cyprus by 90%, Malta by 95%, *p* ≤ 0.01, respectively) in both countries as the vaccination roll-outs progressed, covering about 58.93% of the Maltese population, while Cyprus had fully inoculated about 38.03% of its population. The vaccine strategies and vaccination rates were similar for both countries; yet Malta had the fastest vaccine roll-out. Reluctancy to get vaccinated, significant differences in the vaccination appointment scheduling system, and the freedom of vaccination choice for the citizens in Cyprus may have contributed to a delayed vaccination roll-out. These potential contributing factors should be acknowledged and considered for future vaccination programs and potential COVID-19 boosters.

## 1. Introduction

Vaccination against COVID-19 has been a global priority from the start of the pandemic, with the end of December 2020 seeing the first mRNA vaccine being approved for population inoculation by the European Medical Agency (EMA) [1]. To date (November 2021), four COVID-19 vaccines have been approved by the EMA and are available for purchase by European Union (EU) member states [2]. Although it is accepted that a mass vaccination strategy is the long-term solution to controlling the spread of COVID-19, different European countries have committed to different vaccination strategies and roll-outs, leading to variable populations inoculation speeds [3,4,5]. Regardless, other factors affect a successful mass vaccination strategy, including social attitudes and hesitancy, governance, infrastructure, and religious and ideological factors [6,7].

Two European Island states, the Republic of Cyprus and Malta, are both situated within the Mediterranean Sea and share similar population characteristics, including a total population of less than 1 million (Cyprus 875,900 and Malta 493,559). During the first and second COVID-19 waves, both states were reported to have instituted similar mitigation and restriction strategies (with some exceptions), although differences in COVID-19 positivity rate were noted [8]. Considering these similarities, it was envisaged that a similar vaccination strategy and roll-out speed would be adopted in these two islands. The set timeline was the first six months following the introduction of the vaccines.

However, up until 13 June 2021, Malta had been the leading country in Europe in terms of the highest population proportion fully vaccinated (58.93%), while Cyprus had fully inoculated about a third (38.03%) of its total adult population (18+) [9]. The aim of this study was to compare the COVID-19 vaccination strategies, vaccination rate, and COVID-19 outcome; and at the same time identify and evaluate any similarities or differences between the two small island states of the EU: Cyprus and Malta.

## 2. Materials and Methods

We sought to investigate the extent to which the COVID-19 vaccination strategies, inoculation rate, and COVID-19 outcome differ between the Cyprus and Malta. The aim of this research work was to do a comparison between public health actions and outcomes, as outlined above, to map any significant differences based on these results and potentially inform future public health planning of these, or similar small-size, countries. A comprehensive summary of the vaccination operations and strategy was provided by the ‘COVID-19 Vaccination Logistics’ officer of Malta and the Ministry of Health of Cyprus [10]. Data on COVID-19 infection cases and vaccination were obtained from Ministry of Health websites and COVID-19 dashboards of the two respective Island states in Europe [11,12]. Local published studies, press releases, and newspaper articles were also considered. Vaccination data were obtained from the European Centre for Disease Prevention and Control (ECDC) up until the week 23 of the year 2021 [13].

Comparative assessments were performed between the two countries vaccination strategies, administered vaccine doses, and COVID-19 outcomes in terms of positivity rate and mortality. Microsoft^®^ Excel for Mac Version 16.54 was used to work out all statistical analyses. For the statistical significance tests, we obtained the p-values through t-tests of two sample of unequal variance for the Maltese vs. Cypriot data. Positivity rate is measured by subdividing the daily positive cases by the daily swabs test performed [14]. The COVID-19 period was divided into two phases: (i) from March 2020 until 14 January 2021 represented the pre-vaccination period, and (ii) from 15 January 2021 until 17 June 2021 represented 15 days following the first dose administration in the selected population onwards, i.e., the vaccination period. The case–fatality ratio (CFR) for each COVID-19 phase was calculated by subdividing the confirmed positive cases by the confirmed mortality cases and then multiplying by 100 [15]. This was done to evaluate whether vaccination influenced the mortality in each country. We also identified any lessons learnt that can aid other countries in their vaccination roll-out, as well as inform future vaccine booster coverage protocols.

## 3. Results

### 3.1. The COVID-19 Situation

In Cyprus, 8349 per 100,000 positive cases werebeen reported up until 17 June 2021; in Malta, the number of positive cases reported was 6197 per 100,000. The mortality was 35 per 100,000 for Cyprus and 71 per 100,000 for Malta, as shown in Figure 1. Figure 1 compares the 7-day moving average for mortality and positivity rate from 6 November 2021 to 17 June 2021 for Cyprus (blue colors) and Malta (orange colors), along with the associated restrictions introduced to mitigate the spread. The positivity rate and mortality between December and June significantly differed between the two countries (average positivity rate Cyprus 1.34, Malta 3.37 *p* ≤ 0.01; average mortality Cyprus 7.29, Malta 9.68 *p* ≤ 0.01). Both the positivity rate and mortality for Cyprus declined due to strict public health measures and vaccination roll-out in early January (positivity rate by 95% and mortality by 58%). In contrast, for Malta, there was a sharp increase in positivity rate (64% *p* ≤ 0.01) amidst the vaccination roll-out, as almost no public health restrictions were in place during the Christmas and Carnival period until March, when a lockdown was re-introduced. A distinctive difference in positivity rate (14 per 100,000 population; *p* ≤ 0.01) can also be observed between January to mid-April 2021 between Cyprus and Malta. However, from April onwards it is evident that the positivity rate and mortality decline (positivity rate Cyprus by 82%, Malta by 95%; mortality Cyprus by 90%, Malta by 95%, *p* ≤ 0.01, respectively) in both countries as the vaccination roll-outs progressively covered about 58.93% of the Maltese population, while Cyprus had fully inoculated about 38.03% of its population.

However, the case–fatality ratio (CFR) for Cyprus was lower than that of Malta for both phases, i.e., pre-vaccination phase (until 14 January 2021): Cyprus −0.70; Malta 1.56. During the vaccination phase (15 January to 17 June 2021), the CFR for Cyprus was 0.41 and for Malta was 1.19. Both islands had a decrease in CFR during the vaccination phase.

### 3.2. COVID-19 Vaccines

Cyprus and Malta formed part of the European Union advanced purchase agreement (APA). Both started to receive the vaccine doses in instalments, as per APA agreement, and following vaccine approval by the EMA [2,16]. The first vaccine to arrive in both islands was that of Pfizer BioTech (Comirnaty^®^) on 26 December 2020 [17]. A slight variation in the first availability of the other three vaccines was noted between Cyprus and Malta; with Moderna available in Malta as of 10 January 2021, compared to 4 February 2021 in Cyprus. The AstraZeneca (Vaxzevria^®^) vaccine was available on the 10 February 2021 in Cyprus and the 14 February in Malta 2021; while the Johnson & Johnson vaccine (Janssen^®^) was available on the 14 April 2021 in Cyprus and 6 May 2021 in Malta [18]. Up until 13 June 2021, most of the eligible population for vaccination in both Cyprus and Malta had been inoculated with the Pfizer BioTech vaccine. In Cyprus, 60.13% out of a total of 414,474 first doses and 70.94% out of a total of 277,539 second vaccination doses were of the Pfizer BioTech vaccine. Similarly, in Malta, 66.45% of the 200,073 first doses and 71.75% of the 182,922 second doses werethe Pfizer BioTech vaccine. A similar trend could be observed on age group stratification, as shown in Table 1.

### 3.3. COVID-19 Vaccination Roll-Out Strategies

Difference in vaccination roll-out speed was evident as weeks progressed, as shown in Figure 2, with Malta havingthe fastest vaccination roll-out country across the European Union [5]. Indeed, although the initiation date of vaccination for both islands was the same, a progressive discrepancy in cumulative vaccination population coverage could be observed across the weeks for both the first doses and the second doses (Figure 2). This is also observed on age-group stratification, as shown in Figure 3A (first dose) and Figure 3B (fully vaccinated). Even though both islands prioritized vaccination for the healthcare workers and the elderly (85+ years), differences were seen in the vaccination appointment scheduling systems and its opening and availability to the different age groups, as shown in Table 2. In Cyprus, a vaccination program software was set up, enabling eligible individuals (according to the priority given for different age groups) to schedule a vaccination appointment, with a choice of facility, i.e., the vaccination center, and the vaccine brand, if in stock. An appointment was scheduled by visiting the government portal and the details of the date and time was communicated through text message or phone call. In Malta, the 60+ age groups received a personal invitation by post to their home address with an appointment date and time for both vaccine doses. Individuals belonging to younger age groups below 60 years, were invited to register their interest through different platforms according to the individual’s preference. Eligible individuals could: (i) send a text message with their personal national identification number, (ii) call a designated number, (iii) register through a website, or (iv) register through a mobile application. Those opting for the latter two options were able to choose their preferred time and vaccination center or facility to have the vaccine. A marked difference between the two countries is that in Malta there was no free choice provided to citizens in relation to the vaccine type and brand.

It is noteworthy that themajor differences in the vaccination roll-out between the islands were: (i) up until mid-June 2021 Malta had reached 70% population coverage with one dose and, (ii) Malta started to vaccinate children from the age of 12 years onwards, dissimilarly fromthe situation in Cyprus [19,20]. Cyprus aims to reach a similar population coverage by the middle of the 3rd quarter of 2021, having reached 65% already [21].

### 3.4. Vaccination Infrastructure

Multiple vaccination centers were set up across both islands to facilitate the vaccination roll-out. Additionally, both Cypriot general practitioners (GPs) in the Cypriot National General Health System (GeSy) and Maltese private GPs were enrolled to contribute to the roll-out. However, vaccine administrators differed between Cyprus and Malta. In Cyprus, public sector officers (health visitors, nursing officers, and doctors) were entrusted with the administration of the vaccine. In Malta, apart from the public health officers, volunteers (such as St. John’s Ambulance teams), medical students, dental students, and allied health students were also given the opportunity to be part of the vaccination team [22,23]. Furthermore, the government of Malta announced that, as of 21st July, a mobile vaccination clinic wasset up in different prime locations across the islands. This is to accommodate walk-ins to take the vaccine without requiring an appointment. Additionally, two vaccination hubs have been converted to walk-in clinics, one in Malta (Gateway Hall at University of Malta), and the other in Gozo (Conference and Expo Centre), that also offer a walk-in vaccination service [24]. A similar walk-in clinic for COVID-19 was set-up in Cyprus as of the 15th of July by the Cypriot MoH [25]. This could be a great example for other countries to follow, in order to enhance vaccination coverage, and ultimately set a working example that could be transfer in the vaccination roll-out for other infectious diseases, such as the vaccine against measles, mumps, and rubella (MMR), which still receives resistance from parents over autism claims, despite evidence on safety.

## 4. Discussion

A vaccination coverage targeting between 55% and 82% of the population has been recommended [26]. However, with the emergence of new highly transmissible mutations, such as the Delta variant, this target for vaccination coverage may need to be increased, as it is evident that vaccination hinders hospitalization admissions and infectivity [27]. In both islands, a decrease in positivity and mortality rates was observed when a substantial share of the population werevaccinated, even with just one vaccine dose. Regardless, vaccine hesitancy across the population is still present, which may be one of the contributing factors to the slower vaccination roll-out in Cyprus compared to Malta [28]. On comparing the vaccination strategy across the two islands, Cyprus had a progressive earlier population call for vaccination appointments by age groups (above 30 years) than Malta. Yet, the actual proportion of the population vaccinated by age group was much higher in Malta than Cyprus, suggesting the presence of vaccine hesitancy among the Cypriots. A recent study conducted among healthcare professionals in Malta established a low vaccine hesitancy especially for COVID-19 vaccine, compared to the annual seasonal influenza shot [29].

Another factor that might have hindered the Cypriot vaccination roll-out was the online vaccination registration process. Such a method might have inhibited individuals without internet facilities or who were computer-illiterate from applying, resulting in a low vaccination registration. This was not the case in Malta, where a different system was adopted. Even though individuals below the age of 60 years were asked to register their vaccination interest, a multi-optional portal was established, taking in consideration the various socio-economic strata of the population. The option to choose the vaccine brand might have also led to a delayed vaccination roll-out in Cyprus, especially when the rare thromboembolic side effect of the AstraZeneca vaccine was announced, creating a huge demand for the Pfizer vaccine which exceeded its supply. Although during the same period some vaccine hesitance was noted in Malta, it did not have significant effect on the national vaccination roll-out. Malta, unlike other European countries, did not halt the vaccination roll-out following the reports of the rare AstraZeneca side effects, continually following EMA’s advice [30,31]. An additional factor that might have played a part in the success of Malta’s COVID-19 vaccination was the continuous opening of new vaccination hubs to maintain rapid vaccination of both the first and second doses simultaneously, while maximizing the vaccination orders from all manufacturers.

Although a rapid and high vaccination coverage, with achievement of herd immunity is on every country’s agenda, it does not preclude another COVID-19 wave. Increased population mobility decreased adherence to mitigation measures, such as social distancing and mask wearing, as well as a waning of vaccination protection are all potential contributors to such an occurrence. This is supported by a number of modeling studies that have concluded that an increase in population contact rates may counteract the benefits of a successful vaccination program [32,33,34]. It is therefore important that continuous communication thatencourages vigilance in the population to physical distancing behavior isset in place, along with public health surveillance to monitor the COVID-19 situation and implement timely restrictions, should the need arise.

Several limitations need to be acknowledged for this study. The COVID-19 data and analyses were based on the available data from online sources. The mortality data attributed to COVID-19 does not distinguish between individuals dying with as opposed to dying due to the condition, although these were initially disclosed for Cyprus, during the first wave of the pandemic. There might be other factors than the observed differences in the vaccination roll-out strategies or the similarities of the vaccination uptake at the different rates in the population of the two islands, that can justify for ongoing public health policies and vaccination planning. Therefore, the findings from this study do not necessarily reflect nor can be inferred to explain for the subsequent epidemiological situation and discrepancy between the two islands.

This study highlights a number of factors that potentially contributed to the difference in vaccine uptake and completion, between Malta and Cyprus. However, future research could help the determination of the relative importance of each of these factors, in governing the vaccine uptake statistics. In addition to this, there may be other relevant factors, for example cultural characteristics or socio-economic dissimilarities between the two countries, or even within countries, such as district differences, that can be explored in a qualitative or ethnographic research. The potential benefit of such complementary research is that it may provide an insight on the public health response to vaccination or non-pharmaceutical preventive measures that can justify these differences between the countries.

## 5. Conclusions

Differences in vaccination roll-out speeds were evident between Cyprus and Malta even though these two islands share similar demographic characteristics, formed part of the European Union advanced vaccination purchase agreement, and instituted similar vaccine strategies. Vaccine hesitancy, differences in vaccine registration system, and the individualized option to choose the vaccine brand might have led to a delayed vaccination roll-out in Cyprus.

The freedom of vaccine choice, except in special cases, that was available in Cyprus may have contributed to further confusion around vaccination efficacy and safety amongst the provided brands. These potential contributing factors should be acknowledged by public health officials and considered for future vaccination programs and potential COVID-19 boosters. It is important to note what worked well in Malta, that can inform ongoing and future vaccination policies and campaigns to protect public health, not only for Cyprus but other similar states or small countries. Another key learning point is Malta’s approach of opening early many vaccination centers in the vaccination program, which was particularly helpful in reaching out to all parts of the country and sections of the population. Moreover, the use of many methods in scheduling appointments seems to be a good practice in order toresponding to the computer and digital literacy needs and competencies of all the strata of the population. This could be flagged as a good practice that potentially may release barriers which may create more vaccine reluctancy.

## Figures and Tables

**Figure 1 healthcare-10-00222-f001:**
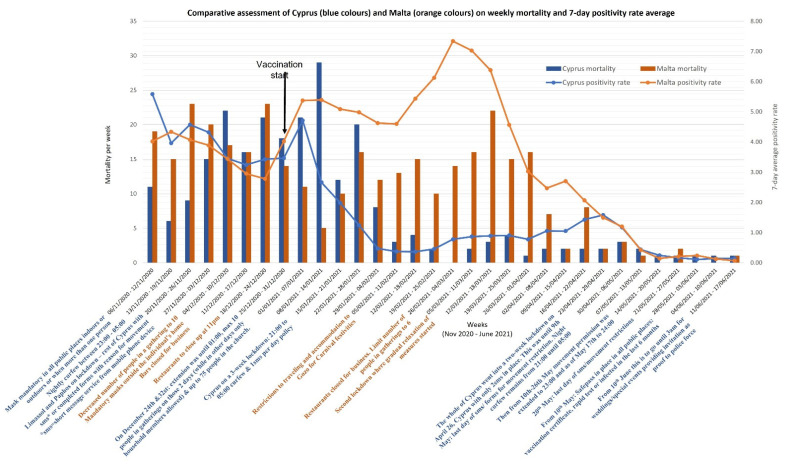
Comparative assessment of the cumulative vaccination population coverage in Cyprus and Malta across weeks according to the different doses. Black arrows represent the proportion difference between both islands for the same week.

**Figure 2 healthcare-10-00222-f002:**
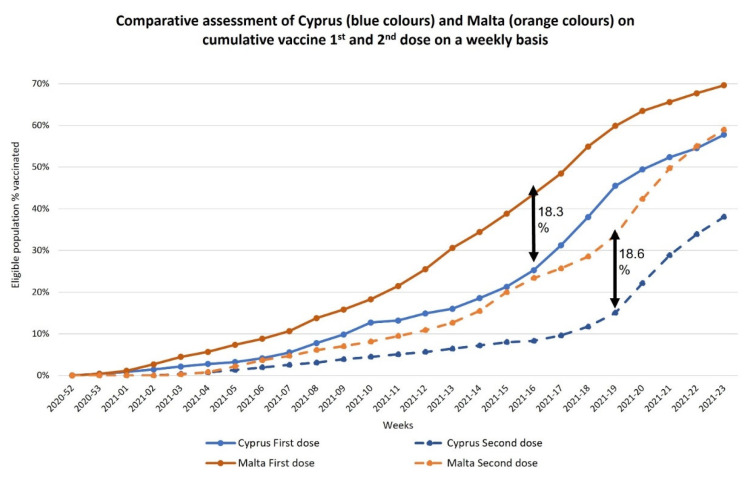
Comparative assessment of the cumulative vaccination population coverage in Cyprus and Malta across weeks according to the different doses.

**Figure 3 healthcare-10-00222-f003:**
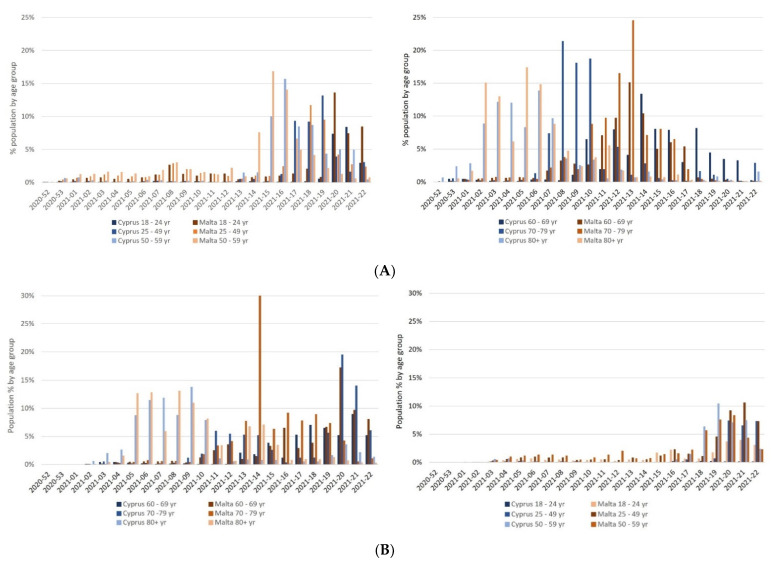
(**A**) Comparative assessment of the weekly first dose vaccination coverage by age groups in Cyprus and Malta. (**B**) Comparative assessment of the weekly full vaccination coverage by age groups in Cyprus and Malta.

**Table 1 healthcare-10-00222-t001:** Comparative analyses of the first dose and fully vaccinated distribution in Cyprus and Malta by age groups and vaccine brand up until 13 June 2021.

		Cyprus		Malta		Cyprus		Malta	
		1st Dose		1st Dose		Fully Vaccinated		Fully Vaccinated	
		N	%*	N	%*	N	%*	N	%*
18–24 years	Pfizer	12,878	15.18%	14,749	35.00%	376	0.44%	5513	13.08%
	Moderna	3477	4.10%	2177	5.17%	44	0.05%	158	0.37%
	AZ	1153	1.36%	3747	8.89%	317	0.37%	3609	8.57%
	J&J					26	0.03%	53	0.13%
	Total	17,508	20.63%	20,673	49.06%	763	0.90%	9333	22.15%
25–49 years	Pfizer	89,889	27.21%	80,926	39.45%	71,277	21.58%	68,505	33.39%
	Moderna	16,334	4.95%	7756	3.78%	3227	0.98%	2125	1.04%
	AZ	34,304	10.39%	30,826	15.03%	2799	0.85%	21,455	10.46%
	J&J					5211	1.58%	170	0.08%
	Total	140,527	42.55%	119,508	58.25%	82,514	24.98%	92,255	44.97%
50–59 years	Pfizer	43,972	40.85%	20,942	35.39%	38,122	35.42%	20,812	35.17%
	Moderna	5215	4.85%	1433	2.42%	954	0.89%	557	0.94%
	AZ	19,743	18.34%	20,960	35.42%	1434	1.33%	5644	9.54%
	J&J					37	0.03%	26	0.04%
	Total	68,930	64.04%	43,335	73.24%	40,547	37.67%	27,039	45.70%
60–69 years	Pfizer	35,071	37.03%	18,503	31.07%	31,378	33.13%	19,306	32.42%
	Moderna	6378	6.73%	2876	4.83%	4268	4.51%	4560	7.66%
	AZ	30,048	31.72%	22,861	38.39%	10,503	11.09%	18,311	30.75%
	J&J					285	0.30%	5	0.01%
	Total	71,497	75.49%	44,240	74.29%	46,434	49.03%	42,182	70.83%
70–79 years	Pfizer	20,878	31.43%	36,209	80.61%	18,729	28.19%	36,847	82.03%
	Moderna	3427	5.16%	4828	10.75%	3364	5.06%	5517	12.28%
	AZ	33,305	50.14%	1673	3.72%	29,020	43.69%	485	1.08%
	J&J					47	0.07%	3	0.01%
	Total	57,610	86.72%	42,710	95.08%	51,160	77.01%	42,852	95.40%
80+ years	Pfizer	24,479	73.01%	21,543	99.33%	23,629	70.47%	19,797	91.28%
	Moderna	824	2.46%	42	0.19%	808	2.41%	53	0.24%
	AZ	4307	12.85%	34	0.16%	2832	8.45%	56	0.26%
	J&J					2	0.01%	1	0.00%
	Total	29,610	88.31%	21,619	99.68%	27,271	81.33%	19,907	91.78%

%*: percentage of eligible population by age group.

**Table 2 healthcare-10-00222-t002:** Comparisons between Cyprus and Malta for the initiation of vaccination appointment by age groups.

	Priority Groups Schedule	
Initiation of Vaccination Appointments	Cyprus	Malta
27 December 2020	Healthcare workers	Healthcare workers and long-term care facility
11 January 2021		Persons living in long-term care facilities—elderly and mental health, and 85+ years
27 January 2021	88+ years	
29 January 2021	86+ years	
1 February 2021	84+ years	All other frontliners and 80–85 years
3 February 2021	83+ years	
8 February 2021		Vulnerable population *
9 February 2021	75–79 years	
17 February 2021	74+ years	
23 February 2021	71+ years	
24 February 2021		Staff at schools and child-care centers
1 March 2021		70–80 years
3 March 2021	69+ years	
6 March 2021		60+ years
9 March 2021	67+ years	
20 March 2021	66+ years	
29 March 2021	Diabetes and Obese	
1 April 2021	64+ years	
7 April 2021	61–63 years	
9 April 2021	59–60 years	
10 April 2021	57–58 years	50+ years
11 April 2021	Specific vulnerable groups	
13 April 2021	55–56 years	
16April 2021	53–54 years	
19 April 2021	Bedridden patients	
21 April 2021	51–52 years	
23 April 2021		40+ years
26 April 2021	43–44 years	
27 April 2021	42 years	
28 April 2021	41 years	
29 April 2021	39–40 years	
4 May 2021	37–38 years	
5 May 2021		30 + years
6 May 2021	35–36 years	
7 May 2021	33–34 years	
9 May 2021	31–32 years	
10 May 2021	29–30 years	
11 May 2021	27–28 years	
12 May 2021	25–26 years	
19 May 2021		16+ years
7 June 2021	23–27 years	
9 June 2021	18–22 years	
16 June 2021		12+ years
18 June 2021	18+ years	

* insulin-dependent diabetics; immunosuppressed; cancer patients undergoing chemotherapy; people treated for cancer in the last six month; patients on dialysis; those admitted to hospital for respiratory problems; patients suffering from cardiac disease or who attend the heart failure clinic; people with Down's syndrome; people who use a BiPap machine.

## Data Availability

All data were collected from publicly available and referenced re-sources. The performed analysis are stored as archives by the co-authors and any researcher interested in the conducted analyses can contact the team directly to access any data.

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
