# Peer review of "Same Pandemic Yet Different COVID-19 Vaccination Roll-Out Rates in Two Small European Islands: A Comparison between Cyprus and Malta"

_healthcare, 2022, doi:10.3390/healthcare10020222_

Round 1

Reviewer 1 Report

If the authors could update the data to the most recent data, that would be more helpful for other countries as the reference.

Reviewer 2 Report

The article "Same pandemic yet different COVID-19 vaccination roll-out 2 rates in two small European islands: a comparison between Cyprus and Malta" is interesting and the subject is important during COVID era. The text is fine, conclusions are good and adequate to to the results. However I have major concern about statistics. In general there is lack of statistic in my opinion. The data are presented only in percentage number without confidence intervals. Fig. 1, Fig. 2, Fig. 3a and 3b, Tab. 1 needs confidence intervals. The same situation in whole text of manuscript such us in line 192 - percentage require confidence intervals. The raw data are not enough. It require statistic. In addition there are to many text in Fig. 1 making it illegible. From minor concern lines 80-82 looks like be not part of the article and should be removed.

Sincerely,

Reviewer

Reviewer 3 Report

Mass vaccination with the Covid-19 vaccine, achieving high population coverage among the total population is arguably the single most important component of long-term control of the pandemic. This paper aims to study the vaccination rates, and in parallel the incidence and case fatality rates Covid-19, in two European island states, which have similar geographic locations, and population characteristics, and participated in the vaccination programme rolled out in the EU.

Although the vaccination strategies were similar in the two countries, Malta had a faster vaccination completion rate than Cyprus. In both countries the incidence and mortality rate from Covid 19 declined, in parallel with the vaccination programme.

The paper tries to analyse reasons for the difference in vaccine uptake between the two countries. They conclude that the employment of a variety of methods for registering and scheduling vaccine appointments in Malta ,compared to the obligatory online registering system in Cyprus, the restriction of the choice of vaccine to be received, in Malta in contrast to freedom of choice in Cyprus, and the potentially greater vaccine hesitancy in the latter country, may be explanations for the difference in vaccine uptake.

Analysis of operational and sociological factors, that modulate success of the vaccination programme is important for configuring future public health measures to prevent infectious diseases. 

Comments to the authors:

  1. This study highlights a number of factors that potentially contributed to the difference in vaccine uptake and completion, between Malta and Cyprus. However, the methodologies used do not enable the determination of the relative importance of each of these factors, in governing the vaccine uptake statistics. In addition to this there may be other relevant factors that the methodology could not identify, for example cultural differences. Therefore, in my opinion the paper should document the difference in the rate of vaccine uptake, and the dynamics of Covid-19 infection rate, and case fatality rate, and this would be a useful contribution.
  2. The factors that contributed to these differences can be addressed in the discussion section with the conclusion being that further well-designed investigation of the relevance of these factors is important and should be pursued.

Round 2

Reviewer 2 Report

I have no more suggestions.

Author Response

Thanks for your work on reviewing our manuscript.

Reviewer 3 Report

The authors have satisfactorily addressed the comments of the reviewer.

Author Response

(The authors gave the same response as above.)
